# Correlation between clinical presentations, radiological findings and high risk histopathological features of primary enucleated eyes with advanced retinoblastoma at Queen Sirikit National Institute of Child Health: 5 years result

**Supawan Surukrattanaskul, Bungornrat Keyurapan, Nutsuchar Wangtiraumnuay** *

Department of Ophthalmology, Queen Sirikit National Institute of Child Health, Bangkok, Thailand

* n.wangtiraumnuay@gmail.com

## Abstract

### Purpose

To evaluate the correlation between clinical presentations, radiological findings and high risk histopathological features of primary enucleated eyes in patients with advanced retinoblastoma at a tertiary child hospital in Thailand.

### Materials and methods

We retrospectively reviewed the medical records of patients who were treated with primary enucleation of tumor eyes between 2015–2020. Demographic data, radiological assessment, and histopathological findings were collected. The association between clinical presentations and high-risk histopathological features in the primary enucleated eyes were evaluated. The radiological findings, which showed characteristic of high risk features, were compared with the histopathological findings.

### Results

Thirty-three enucleated eyes were enrolled in this study. The mean age at diagnosis was 23.12 months. Most patients had unilateral group E retinoblastoma, with no difference in sex and laterality of the eye. Leukocoria was the most common presentation, followed by proptosis and uveitis. Older age at presentation were statistically associated with post laminar cribrosa optic nerve invasion (P-value 0.0027) and high-risk histopathological features in enucleated eyes (P-value 0.0032). Clinical presentations with proptosis were statistically associated with post laminar cribrosa optic nerve invasion, while leukocoria and uveitis were statistically associated with anterior segment invasion. Unifocal intraocular mass with necrosis was the most common histopathological finding. High-risk features were found in 45% of primary enucleated eye. The sensitivity and specificity of magnetic resonance imaging

available upon request from an Institutional Data Access / Ethics Committee (contact via ethic. qsnich@gmail.com) for researchers who meet the criteria for access to confidential data.

**Funding:** The authors received no specific funding for this work.

**Competing interests:** The authors have declared that no competing interests exist.

(MRI) for detected optic nerve invasion in retinoblastoma patients were 75% and 54%, respectively.

## Conclusion

Patients with unilateral retinoblastoma who presented with older age related to high-risk features after enucleation. Ophthalmic examination with slitlamp is the best way for detection of anterior segment invasion. Choroidal invasion was unable to predict with clinical presentation. MRI was the better imaging for detection of post laminar cribrosa optic nerve invasion.

## Introduction

Retinoblastoma is the most common primary intraocular tumor during childhood in Thailand and the rest of the world. Approximately new cases occur at the rate of 9,000 cases per year globally with a 40% burden in the Asia-Pacific region [1]. The incidence of retinoblastoma in Thailand was 3.1/million from the Thai Pediatric Oncology Group (ThaiPOG) [2] and 4.3 per million from three population-based cancer registries, with an incidence trend gradually increasing by 2% annually [3]. In Thailand, the overall survival rate varies from 60% to 94% depending on hospital-based or institute-based in each region of the country [2–4]. Currently, the main problem is that delayed diagnosis leads to advanced stages of the disease. The Global Retinoblastoma Study Group show that the median age at diagnosis in low to middle-income countries (30.5 months) was higher compared to the high-income countries (14.1 months). Low -to middle-income countries also have a higher proportion of locally advanced disease and distant metastasis. The most common presentations at the hospital were leukocoria, strabismus, and proptosis. However, the clinical presentations at the referral center were proptosis, swollen eyelids, red eyes, and orbital cellulitis that were related to the advanced stage of the disease [5]. In two institute-based studies in Thailand, the most common clinical presentation was leukocoria (70%), followed by strabismus, and most patients were staged in group E of the International Classification of Retinoblastoma [4, 6].

The diagnosis of retinoblastoma requires comprehensive ophthalmic evaluation by an ophthalmologist and ophthalmic B-scan ultrasonography related to the history of illness and clinical symptoms. The differential diagnosis of retinoblastoma includes retinal disease, Coats disease, Persistent fetal vasculature, retinal astrocytic hamartoma, and retinopathy of prematurity that usually present the same as leukocoria. Calcification is the most differentiating feature of retinoblastoma, which is evident on ultrasonography. Imaging modalities such as magnetic resonance imaging (MRI) and computed tomography (CT) of the brain and orbit were used to assess tumor characteristics, extraocular extension, and intracranial tumor involvement. MRI is currently recommended because of superior soft-tissue contrast and reduced exposure to ionizing radiation in CT [7]. The classification and staging system of intraocular retinoblastoma is currently defined as a successful treatment by intravenous chemotherapy. Shield and colleagues developed the Intraocular Classification of Retinoblastoma (ICRB) with predicted success outcomes of systemic chemotherapy. The eyes in groups A–C had a globe salvage rate of more than 90%. The globe salvage rate of group D was 47%, and group E was enucleated and excluded from the studies [8, 9]. The treatment modality depends on disease staging, laterality, tumor location, and parent acceptance. The ICRB is shown in Table 1.

Unilateral group E and some group D retinoblastomas usually have no visual prognosis due to severe and irreversible anatomical damage of the eye, which is recommended to enucleate

**Table 1. Intraocular Classification of Retinoblastoma (ICRB).**

| Group A | Retinoblastoma ≤ 3 mm (in basal dimension or thickness) |
|---|---|
| Group B | Retinoblastoma > 3 mm (in basal dimension or thickness) or<br>• Macular location (≤3 mm to foveola)<br>• Juxtapapillary location (≤1.5 mm to disc)<br>• Additional subretinal fluid (≤3 mm from margin) |
| Group C | Retinoblastoma with:<br>• Subretinal seeds ≤ 3 mm from tumour<br>• Vitreous seeds ≤ 3 mm from tumour<br>• Both subretinal and vitreous seeds ≤ 3 mm from tumour |
| Group D | Retinoblastoma with:<br>• Subretinal seeds > 3 mm from tumour<br>• Vitreous seeds > 3 mm from tumour<br>• Both subretinal and vitreous seeds > 3 mm from retinoblastoma |
| Group E | • Extensive retinoblastoma occupying >50% globe or with<br>• Neovascular glaucoma<br>• Opaque media from haemorrhage in anterior chamber, vitreous or subretinal space<br>• Invasion of postlaminar optic nerve,<br>• choroid (>2 mm), sclera, orbit, anterior chamber |

the tumor eye. Histopathologic findings in enucleated eyes will report growth pattern, tumor cell differentiation, rosette formation, optic nerve invasion, and resection margin. The risk of tumor recurrence and distant metastasis are considered high-risk features and require adjuvant chemotherapy. High-risk features of histological findings were defined as the presence of anterior segment invasion (anterior chamber seeding, iris infiltration, ciliary muscle/body infiltration), massive (≥3 mm) choroidal invasion, retrolaminar optic nerve invasion, invasion of the optic nerve surgical margin, combined non-massive choroidal and prelaminar/laminar optic nerve invasion, and scleral/extra scleral infiltration [10–12].

The Queen Sirikit National Institute of Child Health, a tertiary referral hospital in Bangkok, provides retinoblastoma treatment with local cryotherapy, transpupillary thermotherapy, systemic chemotherapy, and primary enucleation in the advanced stage of retinoblastoma. In patients with bilateral advance disease or requisitioning of globe salvage treatment, we refer to other institute-based hospitals for other treatment modalities. Therefore, our study aimed to evaluate the clinical presentation, radiological findings, histopathology results, and frequency of high-risk features in eyes with primary enucleated group E retinoblastoma and analyze the relationship between clinical and radiological characteristics associated with high-risk features.

## Methods

The medical records of all retinoblastoma patients who underwent primary enucleation at the Queen Sirikit National Institute of Child Health from January 2015 to January 2020 were retrospectively reviewed. Enucleated eyes with pathological confirmation of other diseases, such as Coats disease and eyes with previous surgical manipulation were excluded. Ethical approval was obtained from the Research Ethics Review Committee of Queen Sirikit National Institute of Child Health (Children's Hospital).

Data were collected for demographic characteristics, clinical presentations, radiology findings of CT or MRI brain with orbit, and histopathological reports. All pathological reports were performed at the Institute of Pathology, Department of Medical Services, Ministry of Public Health, Bangkok. All slides were reviewed for tumor characteristics, tumor differentiation, and the presence of high-risk histopathological features. Tumor differentiation was categorized as differentiated (Rosettes, Flexner–Wintersteiner rosettes, Homer Wright rosettes,

Fleurettes, Rosettes), undifferentiated, and poorly differentiated. High-risk features were defined as the presence of anterior chamber seeding, iris infiltration, ciliary muscle infiltration, massive (≥3 mm) choroidal invasion, retrolaminar optic nerve invasion, invasion of the optic nerve at the surgical margin, combined non-massive choroidal and prelaminar/laminar optic nerve invasion, and scleral/extra scleral infiltration.

Data are presented as counts and percentages for categorical data. The data were reported as means and standard deviations. A nonparametric test, Mann–Whitney U test, and Fisher's exact test were used to evaluate the relationship between clinical presentations and high-risk histopathological features. The association analysis was performed with STATA 16.0 version (StataCorp, Texus, USA). Association between categorical variables were analyzed by Fisher exact test. For comparing continuous variables between two groups, Mann whitney U test was performed. P value of less than 0.05 was considered significant. The sensitivity and specificity of the MRI for prediction of the optic nerve involvement, choroidal invasion and anterior segment involvement in retinoblastoma were calculated.

## Results

All patients underwent fundus examination and ophthalmic ultrasonology. Most of patients had radiological assessment under anesthesia before the provisional diagnosis of retinoblastoma. According to ICRB classification, after clinical diagnosis and staging of the disease to group E retinoblastoma. The information both glove salvage regimen or primary enucleation were provided to parents for consensus on treatment. All thirty-three primary enucleated eyes were enrolled in this study. The mean age during diagnosis was 23.12 months (SD±14.70). Ninety-one percent of patients had unilateral retinoblastoma and no significant difference in sex or laterality of the eye was observed. Three patients with bilateral retinoblastoma were Group A in one eye and Group E in the following eye. In Group A, retinoblastoma eyes were treated with local transpupillary thermotherapy, cryotherapy, and primary enucleation of the advanced eye. After enucleation, all bilateral patients received systemic chemotherapy monthly and closed follow-up fundus examination every 2 months, with no evidence of recurrence in the remaining eye. Leukocoria was the most common presentation, followed by proptosis, uveitis, and strabismus. Most patients have a combination of leukocoria and other symptoms of advanced diseases, such as proptosis, pseudohypopeons, and orbital cellulitis. The demographic data and clinical presentation are shown in Table 2. The intraocular pressure were measure only in some patients, all the patients with proptosis revealed high intraocular pressure. The neovascular glaucoma was not mentioned in the record.

Ophthalmic ultrasound was performed in all patients. It revealed a hyperechoic mass occupying >50% of the posterior segment of the globes with calcification deposited in the tumor mass. Twenty eight patients (84%) were performed MRI brain with orbit (17 patients) or CT brain with orbit (11 patients) under general anesthesia, there were no identified an intracranial extension. Intraocular calcification of enucleated eyes were reported in all CT scan (100%), but only 7 MRI scans (41%). The optic nerve involvement were not mentioned in all CT scans, but 3 out of 11 negative CT patients had a pathologically confirmed post-laminar invasion. For MRI, the optic nerve involvement were reported on 9 scans (53%), but only 3 out of 9 MRI patients had a pathological post-laminar invasion. An extraocular extension was found on one MRI scan, but histopathology reported tumor necrosis of more than 90% with pre-laminar optic nerve invasion, and the resection margin was free of tumor. Anterior segment invasion and choroidal invasion, each was found on only one MRI scan with compatible with histopathology. Radiological findings in patients with retinoblastoma before primary enucleation are shown in Table 3.

**Table 2. Demographic data and clinical presentation of primary enucleated retinoblastoma patients.**

| Data | Number |
|---|---|
| Age (months) (mean±SD) | Mean 23.12 (±14.70) |
| Sex | |
| • Male | 17 (52%) |
| • Female | 16 (48%) |
| Laterality | |
| • Right eye | 14 (42%) |
| • Left eye | 19 (58%) |
| Unilateral retinoblastoma | 30 (91.%) |
| Bilateral retinoblastoma | 3 (9%) |
| Family history of retinoblastoma | |
| • No | 33 (100%) |
| • Yes | 0 |
| Clinical presentation | |
| • Leukocoria | 30 (91%) |
| • Proptosis | 11 (33%) |
| • Uveitis | 5 (15%) |
| • Strabismus | 4 (12%) |
| • Aseptic orbital cellulitis | 2 (6.%) |
| • Cataract | 2 (6%) |

Gross pathology of tumors is mostly found as a unifocal grey-white friable intraocular mass. The mean±SD maximum diameter of the tumor mass was 18.88±0.32 mm, an estimate involving >80% of globe diameter (normal A-P diameter/vertical/horizontal was 22–23 mm). Tumor necrosis and optic nerve invasion were the most common histopathological findings (81% and 28%, respectively). The optic nerve was invaded posterior to the lamina cribrosa in nine enucleated eyes, and two eyes were not free of tumor at the resection margin. Two patients with tumors at the resection margins (10 and 16 millimeters optic nerve length) do not have evidence of optic nerve invasion or extraocular extension on MRI findings. Microscopic findings reveal that retinal tumor cell differentiation was more than the undifferentiated tumors. The characteristic rosettes of retinoblastoma such as, Flexner–Wintersteiner rosettes (34%), and Homer Wright rosettes (22%) was observed. Fifteen eyes (45%) had high-risk features. There were 4 eyes with anterior segment involvement in histopathological report which compatibled with the clinical ophthalmic examination (hypopyon, anterior chamber cell and anterior chamber mass), the radiological report detected only one anterior segment involvement from MRI scan. The histopathology details are shown in Table 4.

**Table 3. Radiological modality assessment in retinoblastoma patients.**

| Data | Radiological assessment | |
|---|---|---|
| Total Retinoblastoma patients | 33 | |
| MRI assessment (number) | 17 (52%) | |
| CT assessment (number) | 11 (33%) | |
| Radiological finding | **MRI (Total = 17)** | **CT (Total = 11)** |
| • Calcification | 7 (41%) | 11 (100%) |
| • Optic nerve invasion | 9 (53%) | 0 |
| • Vitreous hemorrhage, Subretinal hemorrhage | 4 (24%) | 0 |
| • Enlarge globe | 2 (12%) | 0 |
| • Choroid invasion | 1 (6%) | 0 |
| • Anterior chamber invasion | 1 (6%) | 0 |
| • Extraocular extension | 1 (6%) | 0 |

**Table 4. Pathological finding in primary enucleated retinoblastoma patients.**

| Pathological finding | Number (%) |
|---|---|
| **Gross finding** | |
| • Total (number) | 33 |
| • Size of mass (mean maximal diameters ±SD) | 18.88 ±0.32 mm |
| • Mean of optic nerve length (min-max, mm) | 12.25 mm (3-20mm) |
| **Microscopic finding** | |
| • Total (number) | 32 |
| • High-risk feature | 15 (47%) |
| ▪ Optic nerve invasion: post laminar invasion | 9 (28%) |
| ▪ Anterior chamber invasion | 4 (13%) |
| ▪ Massive choroidal invasion | 3 (9%) |
| ▪ Combined non-massive choroidal & pre laminar invasion | 2 (6%) |
| • Other | 26 (81%) |
| ▪ Necrosis | 7 (22%) |
| ▪ Calcification | 4 (13%) |
| ▪ Vitreous seeding | 3 (9%) |
| ▪ Hemorrhage | |
| **Retinal cell differentiation** | |
| • Total (number) | 32 |
| • Flexner-Wintersteiner rosettes | 11 (34%) |
| • Homer Wright rosettes | 7 (22%) |
| • Poor differentiation | 6 (19%) |
| • Fleurettes | 3 (9%) |
| • Rosettes | 3 (9%) |
| • Undifferentiation | 13 (41%) |

A nonparametric test, Mann–Whitney U test, and Fisher's exact test were used to evaluate the relationship between demographic data, clinical presentations and high-risk histopathological features, and also some findings of high risk features (anterior segment invasion, choroidal invasion, post laminar cribrosa optic nerve invasion) as shown in Table 5. Older age at presentation were statistically associated with high-risk histopathological features (p 0.0032) and post laminar cribrosa optic nerve invasion (p 0.0027). Proptosis were statistically associated with post laminar cribrosa optic nerve invasion (p 0.033). Anterior uveitis and leukocoria were statistically associated with anterior segment invasion (p 0.000 and p 0.033, respectively). The mean age of patients in the high-risk feature group was 31.26±13.80 months, which was higher than that of patients in the non-high-risk feature group 16.29±11.88 months (p = 0.0032).

The sensitivity, specificity, positive predictive value and negative predictive value of the MRI orbit for predict the optic nerve involvement, chroidal invasion and anterior segment invasion were calculated as shown in Table 6.

## Discussion

In this study, the clinical characteristics of retinoblastoma included older age at diagnosis, unilateral advanced disease, and no family history of retinoblastoma. Similar to the Global Retinoblastoma Study Group, the estimated age at presentation in Low-Middle income countries were 24.4 (12.2–37.3 months) with more advanced disease and a smaller proportion of family history of retinoblastoma [5]. Older age at diagnosis explained the stage of disease in our patients. To save the life of a child as a priority, group E according to ICRB classification can have consensus with parents for unilateral primary enucleation. Leukocoria is usually presenting symptom. Public education and ophthalmic screening with red reflex test may help in early detection and avoid enucleation.

Ophthalmic ultrasound was easily performed without sedation, and all of the retinoblastoma patients showed pathognomonic features with a hyperechoic mass occupying >50% of

**Table 5. The correlation between demographic data & clinical presentations and each type of high-risk histopathological features by using Fisher's exact test and Mann–Whitney U test*.**

| Factors | Anterior segment invasion | | | Choroidal invasion > 3 mm | | | Post laminar cribrosa optic nerve invasion | | | Any high-risk features | | |
|---|---|---|---|---|---|---|---|---|---|---|---|---|
| | Yes | No | P value | Yes | No | P value | Yes | No | P value | Yes | No | P value |
| **Demographic data** | | | | | | | | | | | | |
| Mean age (months) | 38.25 | 21.03 | 0.1221* | 31.14 | 20.96 | 0.0610* | 36.11 | 18.25 | **0.0027**\* | 31.60 | 16.06 | **0.0032**\* |
| ±SD (months) | 19.60 | 12.99 | | 14.58 | 14.25 | | 13.08 | 12.27 | | 13.81 | 11.57 | |
| Sex | | | | | | | | | | | | |
| • Male | 2 | 15 | 1.000 | 2 | 15 | 0.225 | 5 | 12 | 1.000 | 7 | 10 | 0.732 |
| • Female | 2 | 14 | | 5 | 11 | | 4 | 12 | | 8 | 8 | |
| **Clinical presentations** | | | | | | | | | | | | |
| Proptosis | | | | | | | | | | | | |
| • Yes | 2 | 9 | 0.586 | 3 | 8 | 0.661 | 6 | 5 | **0.033** | 7 | 4 | 0.163 |
| • No | 2 | 20 | | 4 | 18 | | 3 | 19 | | 8 | 14 | |
| Leukocoria | | | | | | | | | | | | |
| • Yes | 2 | 28 | **0.033** | 7 | 23 | 1.000 | 8 | 22 | 1.000 | 13 | 17 | 0.579 |
| • No | 2 | 1 | | 0 | 3 | | 1 | 2 | | 2 | 1 | |
| Strabismus | | | | | | | | | | | | |
| • Yes | 1 | 3 | 0.420 | 0 | 4 | 0.555 | 1 | 3 | 1.000 | 1 | 3 | 0.607 |
| • No | 3 | 26 | | 7 | 22 | | 8 | 21 | | 14 | 15 | |
| Uveitis | | | | | | | | | | | | |
| • Yes | 4 | 1 | **0.000** | 1 | 4 | 1.000 | 1 | 4 | 1.000 | 4 | 1 | 0.152 |
| • No | 0 | 28 | | 6 | 22 | | 8 | 20 | | 11 | 17 | |

* Mann–Whitney U test

the posterior segment of the globes with calcification deposited in the tumor mass. The time for imaging at our center was 1–3 weeks, and MRI takes more waiting time than CT brain with orbit. Most of our patients came from rural areas in Thailand and were admitted for investigation and enucleation in a single visit, so CT may be performed in some patients. MRI shows superior optic nerve, choroid, and anterior chamber invasion, whereas findings on CT imaging shows superior on intraocular calcified detection. However, the false positive and false negative optic nerve invasion in MRI scans were detected in our study. The optic nerve should be cut as long as possible to avoid residual tumor at resection margin, even negative MRI scan.

Histopathological information in our study has limited data on tumor differentiation categorized into well-differentiated (>50% known as Homer–Wright rosettes) and poorly

**Table 6. The sensitivity and specificity of the MRI for predict the optic nerve involvement in retinoblastoma.**

| Compare of the involvement in MRI and pathology | Involvement by pathology | Not involvement by pathology | Sensitivity | Specificity | Positive predictive value | Negative predictive value |
|---|---|---|---|---|---|---|
| **Optic nerve** | | | 75% | 54% | 33% | 88% |
| Involvement by MRI | 3 | 6 | | | | |
| Not involvement by MRI | 1 | 7 | | | | |
| **Choroid** | | | 25% | 100% | 100% | 81% |
| Involvement by MRI | 1 | 0 | | | | |
| Not involvement by MRI | 3 | 13 | | | | |
| **Anterior chamber** | | | 33% | 100% | 100% | 88% |
| Involvement by MRI | 1 | 0 | | | | |
| Not involvement by MRI | 2 | 14 | | | | |

differentiated (<50% known as Flexner–Wintersteiner rosettes). The most common histo-pathological finding was necrosis. An incidence of high-risk features was 45.45%. The reported incidence of high-risk features in eyes with primary enucleated retinoblastoma varies between 13–36% [13–16]. In ICRB, group E has a high incidence of high-risk features of approximately 39–50%, [13, 14, 16] and choroid, optic nerve invasion was a common feature in approximately 50% of all features. [14, 17]. The risk factor associated with high-risk features in our study was older age at presentation (31.26±13.81 months, *p*-value 0.0032). A previous study showed that older age at presentation, presence of hyphema, pseudohypopyon, and orbital cellulitis were also risk factors for high-risk features. [11] Our study showed the ophthalmic examination with slitlamp was the best way for detection of anterior segment invasion. Choroidal invasion was unable to predict with clinical presentation. The optic nerve transection should be carefully cut as long as possible especially in older children and the patients with proptosis with high intraocular pressure. Radiological assessment should include ophthalmic ultrasonography and MRI of the brain with orbit for differential diagnosis and assessment of extraocular/intracranial extension. The previous study showed the MRI study had sensitivity 40–89% and specificity 72–93% [18–20], while our study reported same level of sensitivity (75%) but lower level of specificity (54%). The false positive in our center is high. However, MRI study is still the modality of choise for retinoblastoma. The adjustment of magnetic field strength and protocols such as higher magnetic field strength (3 T than 1.5 T), performing fat suppression, and thinner slice thickness (< 3 mm) [19] may increase the accuracy of the MRI study in our center. Even though, the MRI revealed 100% specificity and 100% positive predictive value for choroidal invasion and anterior segment involvement, the sample size of our study is too small for making the conclusion.

## Conclusion

Patients with unilateral retinoblastoma who presented with older age related to high-risk features after enucleation. Leukocoria is usually presenting symptom, the public education and ophthalmic screening should be implemented for early detection and avoid enucleation.

Ophthalmic examination with slitlamp is the best way for detection of anterior segment invasion. MRI was the better imaging for detection of post laminar cribrosa optic nerve invasion. The optic nerve transection should be carefully cut as long as possible, even negative MRI scan for avoid residual tumor. The treatment plans include the constantly follow up and the consideration of adjuvant systemic chemotherapy after primary enucleation to reduce the risk of metastasis.

## Supporting information

**S1 Data.**
(XLSX)

## Author Contributions

**Conceptualization:** Supawan Surukrattanaskul.

**Data curation:** Supawan Surukrattanaskul, Bungornrat Keyurapan, Nutsuchar Wangtiraumnuay.

**Formal analysis:** Supawan Surukrattanaskul, Nutsuchar Wangtiraumnuay.

**Investigation:** Nutsuchar Wangtiraumnuay.

**Methodology:** Supawan Surukrattanaskul.

**Project administration:** Supawan Surukrattanaskul, Bungornrat Keyurapan.

**Supervision:** Bungornrat Keyurapan, Nutsuchar Wangtiraumnuay.

**Validation:** Nutsuchar Wangtiraumnuay.

**Visualization:** Nutsuchar Wangtiraumnuay.

**Writing – original draft:** Supawan Surukrattanaskul.

**Writing – review & editing:** Bungornrat Keyurapan, Nutsuchar Wangtiraumnuay.

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
