## [Decision Letter · Decision Letter 0]

27 Apr 2022

PONE-D-22-06589Correlation between Clinical Presentations, Radiological Findings and High Risk Histopathological Features of Primary Enucleated Eyes with Retinoblastoma at Queen Sirikit National Institute of Child Health: 5 years resultPLOS ONE

Dear Dr. Wangtiraumnuay,

Thank you for submitting your manuscript to PLOS ONE. After careful consideration, we feel that it has merit but does not fully meet PLOS ONE’s publication criteria as it currently stands. Therefore, we invite you to submit a revised version of the manuscript that addresses the points raised during the review process.

ACADEMIC EDITOR:The manuscript needs some grammatical and style revision. Some sections of the manuscript are redundant and can trimmed. The authors should also focus in the discussion on the significance of their findings.

We look forward to receiving your revised manuscript.

Kind regards,

Ahmed Awadein, MD, Ph.D, FRCS

Academic Editor

PLOS ONE

Journal Requirements:

Reviewers' comments:

Reviewer's Responses to Questions

**Comments to the Author**

1. Is the manuscript technically sound, and do the data support the conclusions?

Reviewer #1: Partly

Reviewer #2: Yes

2. Has the statistical analysis been performed appropriately and rigorously? 

Reviewer #1: Yes

Reviewer #2: Yes

3. Have the authors made all data underlying the findings in their manuscript fully available?

Reviewer #1: Yes

Reviewer #2: Yes

4. Is the manuscript presented in an intelligible fashion and written in standard English?

Reviewer #1: No

Reviewer #2: Yes

5. Review Comments to the Author

Reviewer #1: Although there are many previous similar reports , which have described the high risk histopathological features after enucleation and yielded more or less similar results and conclusions, your study analyzed added the analysis of the correlation with radiological findings and clinical presentation.

1/3 of your included patients (11 out of 33) did CT ONLY as the imaging modality prior to enucleation. Apart from the fact you already mentioned in your introduction that the high dose of ionizing radiation used in CT should be avoided and carries a high risk in hereditary cases, MRI is definitely the scan of choice for detection of ON, choroid &/or scleral involvement. So you relied on a relatively less accurate test in 33% of your patients for the analysis of the radiological findings and its correlation with HP and clinical findings.

The "Introduction" section is redundant and would be better summarized.

The English in the manuscript is not of publication quality and require major improvement and proof-reading.

Reviewer #2: The manuscript has been drafted with great efforts. I appreciate the hard-work put in by the team.

1. As there was an association seen between age and high-risk factors in the study, it will be interesting to have the p-value for the same as a part of the abstract too.

2. The conclusion section of the abstract has a blend of results and conclusion. It will be great if the authors could segregate the same.

3. I think it will be a better idea to include the word "advanced retinoblastoma" as a part of the title too.

4. The introduction has been written very well with appropriate information for the readers.

5. Line 135-136 - Advanced retinoblastoma term followed by grouping, shall cover the spectrum neatly. Kindly reconsider the terms used here. Or are all the groups included? It is slightly unclear here.

6. Line 137 - Was it a consecutive enrollment? If yes, please mention the same.

7. Line 143 - Was the chemotherapy given monthly or 2-monthly? Kindly highlight.

8. Line 145 - Was it just high IOP or NVG?

9. Table 2 - Inclusion of high IOP along with proptosis seems a little inaccurate. Best to cover NVG separately.

10. Line 153 - Please mention the number. As all the cases weren't group E.

11. Table 3 - It will be interesting if the authors could include the histopathological risk factors in the table and compare with the radiological findings. A reflection of the same gives a higher impact. It has been partly covered in table 6, but it will be best if combined while commenting on all the variables.

12. Line 190 - Highlighting again that proptosis and glaucoma are best if individually assessed and commented upon.

13. Kindly keep the percentage format symmetrical throughout the manuscript i.e. rounded to one decimal or 2 decimals or none.

14. Line 232 - 233 - Kindly cite the reference here.

15. Line 247 - A brief comment on the conclusion of the study is recommended.

6. PLOS authors have the option to publish the peer review history of their article (what does this mean?). If published, this will include your full peer review and any attached files.

Reviewer #1: No

Reviewer #2: No

---

## [Author Response · Author response to Decision Letter 0]

21 May 2022

Response to reviewers’ comments

Reviewer #1: Although there are many previous similar reports , which have described the high risk histopathological features after enucleation and yielded more or less similar results and conclusions, your study analyzed added the analysis of the correlation with radiological findings and clinical presentation.

1/3 of your included patients (11 out of 33) did CT ONLY as the imaging modality prior to enucleation. Apart from the fact you already mentioned in your introduction that the high dose of ionizing radiation used in CT should be avoided and carries a high risk in hereditary cases, MRI is definitely the scan of choice for detection of ON, choroid &/or scleral involvement. So you relied on a relatively less accurate test in 33% of your patients for the analysis of the radiological findings and its correlation with HP and clinical findings. 

Response: The explanation of CT scan was under discussion “Most of our patients came from rural areas in Thailand and were admitted for investigation and enucleation in a single visit, so CT may be performed in some patients.”

The "Introduction" section is redundant and would be better summarized. 

Response: Removed the sentence “The International Intraocular Retinoblastoma Classification (IIRC) classifies retinoblastoma as A–E, depending on tumor size, location, and presence of vitreous seeding and/or retinal detachment.” following reviewer 1 suggestion.

The English in the manuscript is not of publication quality and require major improvement and proof-reading. 

Response: English proof was done by editage website.

Reviewer #2: The manuscript has been drafted with great efforts. I appreciate the hard-work put in by the team.

1. As there was an association seen between age and high-risk factors in the study, it will be interesting to have the p-value for the same as a part of the abstract too.

Response: Added P-value to abstract.

2. The conclusion section of the abstract has a blend of results and conclusion. It will be great if the authors could segregate the same.

Response: Moved the sentence “Unifocal intraocular mass with necrosis was the most common histopathological finding.” and “High-risk features were found in 45% of primary enucleated eye.” from the conclusion to the result section.

3. I think it will be a better idea to include the word "advanced retinoblastoma" as a part of the title too.

Response: Added the word “advanced retinoblastoma to the title”

4. The introduction has been written very well with appropriate information for the readers.

Response: Removed the sentence “The International Intraocular Retinoblastoma Classification (IIRC) classifies retinoblastoma as A–E, depending on tumor size, location, and presence of vitreous seeding and/or retinal detachment.” following reviewer 1 suggestion.

5. Line 135-136 - Advanced retinoblastoma term followed by grouping, shall cover the spectrum neatly. Kindly reconsider the terms used here. Or are all the groups included? It is slightly unclear here.

Response: According to ICRB classification, after clinical diagnosis and staging of the disease to group D or group E retinoblastoma. The globe salvage regimen or primary enucleation were provided as the options.

6. Line 137 - Was it a consecutive enrollment? If yes, please mention the same.

Response: Added “All thirty-three primary enucleated eyes were enrolled in this study.”

7. Line 143 - Was the chemotherapy given monthly or 2-monthly? Kindly highlight.

Response: After enucleation, all bilateral patients received systemic chemotherapy monthly and closed follow-up fundus examination every 2 months.

8. Line 145 - Was it just high IOP or NVG?

Response: The intraocular pressure were measure only in some patients, all the patients with proptosis revealed high intraocular pressure. The neovascular glaucoma was not noted in the record.

9. Table 2 - Inclusion of high IOP along with proptosis seems a little inaccurate. Best to cover NVG separately.

Response: The intraocular pressure were measure only in some patients, all the patients with proptosis revealed high intraocular pressure. The neovascular glaucoma was not noted in the record.

10. Line 153 - Please mention the number. As all the cases weren't group E.

Response: Rechecked the data, all the patients were group E retinoblastoma in one eye. According to ICRB classification, after clinical diagnosis and staging of the disease to group E retinoblastoma. Line 153 was not change. 

11. Table 3 - It will be interesting if the authors could include the histopathological risk factors in the table and compare with the radiological findings. A reflection of the same gives a higher impact. It has been partly covered in table 6, but it will be best if combined while commenting on all the variables.

Response: Change table 6, added choroidal involvement and anterior chamber invasion. And added the discussion “Even though, the MRI revealed 100% specificity and 100% positive predictive value for choroidal invasion and anterior segment involvement, the sample size of this group is too small for making the conclusion.”

12. Line 190 - Highlighting again that proptosis and glaucoma are best if individually assessed and commented upon.

Response: The intraocular pressure were measure only in some patients, all the patients with proptosis revealed high intraocular pressure. The neovascular glaucoma was not noted in the record.

13. Kindly keep the percentage format symmetrical throughout the manuscript i.e. rounded to one decimal or 2 decimals or none.

Response: Keep the percentage format symmetrical throughout the manuscript rounded to none. 

14. Line 232 - 233 - Kindly cite the reference here.

Response: Added reference [11]. 

15. Line 247 - A brief comment on the conclusion of the study is recommended.

Response: Added conclusion.

---

## [Editor Report · Decision Letter 1]

9 Jun 2022

Correlation between clinical presentations, radiological findings and high risk histopathological features of primary enucleated eyes with advanced retinoblastoma at Queen Sirikit National Institute of Child Health: 5 years result

PONE-D-22-06589R1

Dear Dr. Wangtiraumnuay,

We’re pleased to inform you that your manuscript has been judged scientifically suitable for publication and will be formally accepted for publication once it meets all outstanding technical requirements.

Kind regards,

Ahmed Awadein, MD, Ph.D, FRCS

Academic Editor

PLOS ONE
---

## [Editor Report · Acceptance letter]

24 Jun 2022

PONE-D-22-06589R1 

Correlation between clinical presentations, radiological findings and high risk histopathological features of primary enucleated eyes with advanced retinoblastoma at Queen Sirikit National Institute of Child Health: 5 years result 

Dear Dr. Wangtiraumnuay:

I'm pleased to inform you that your manuscript has been deemed suitable for publication in PLOS ONE. Congratulations! Your manuscript is now with our production department. 

Kind regards, 

on behalf of

Dr. Ahmed Awadein 

Academic Editor

PLOS ONE